# Prevalence of *Aeromonas* spp. Infection in Pediatric Patients Hospitalized with Gastroenteritis in Latvia between 2020 and 2021

**DOI:** 10.3390/children9111684

**Published:** 2022-11-02

**Authors:** Irina Grave, Aleksandra Rudzate, Anda Nagle, Edvins Miklasevics, Dace Gardovska

**Affiliations:** 1Riga Stradins University, LV1007 Riga, Latvia; 2Children Clinical University Hospital, Bernu Kliniska Universitates Slimnica, LV1004 Riga, Latvia; 3Institute of Oncology, LV1007 Riga, Latvia; 4Department of Paediatrics, Riga Stradins University, LV1007 Riga, Latvia

**Keywords:** *Aeromonas*, gastroenteritis, virulence, pediatric, antimicrobial resistance

## Abstract

Purpose: *Aeromonas* species are emerging human enteric pathogens. However, there is no systematic analysis of *Aeromonas* infection in the pediatric population in Latvia. The aim of the study was to describe potential sources, prevalence of infection, associated virulence factors and antimicrobial resistance of *Aeromonas* spp. isolated from fecal samples. Methods: Stool samples (n = 1360) were obtained from the Children’s Clinical University Hospital between 2020 and 2021. The target population was pediatric patients, 0 to 18 years of age, with a preliminary diagnosis of gastroenteritis. Identification was performed by Maldi-TOF, antimicrobial resistance by Vitek2 and 9 virulence factors by polymerase chain reaction (PCR). Results: *Aeromonas* spp. were isolated in 50 stool samples; positive findings made up 3.6% of all study cases and included four species: *A. hydrophila*, *A. caviae*, *A. veronii*, and *A. eucrenophila*. In 42% of the samples, *Aeromonas spp.* appeared alongside the other significant pathogens: *Campylobacter jejuni, Salmonella Enteritidis*, *Salmonella Typhimurium*, *Yersinia enterocolitica*, norovirus, adenovirus, and rotavirus. The study population positive for *Aeromonas* spp. infection contained 28 male (56%) and 22 female (44%) patients; median age was 4.56 years. The most common symptoms were: diarrhea, blood in stool, vomiting, abdominal pain, and fever. Aside from expected natural resistance, no significant antibacterial resistance was detected. The presence of multiple virulence genes was noticed in all isolates. No statistically significant correlation was found between the virulence patterns, bacterial species, and the intensity of clinical symptoms. Discussion: According to the clinical data and the results of this study *Aeromonas* spp. has an important role in pediatric practice and requires appropriate attention and monitoring.

## 1. Introduction

The genus *Aeromonas* belongs to the *Aeromonadaceae* family and represents a group of rod-shaped motile Gram-negative, facultative anaerobic bacteria. *Aeromonas* spp. can be found in various aquatic environments including treated drinking water, soil, animals, and food products [1,2,3,4]. Generally, members of the genus are characteristically divided into three biochemically differentiated groups (*Aeromonas hydrophila*, *Aeromonas caviae,* and *Aeromonas sobria*.) The taxonomy of this genus is complex and species can be misidentified when phenotypic identification methods are used. Therefore, its correct identification requires molecular methods [3,5,6].

Some *Aeromonas* species are regarded as disease-causing pathogens in fish and other cold-blooded animals; however, *A. hydrophila, A. caviae, A. dhakensis, A. veronii*, and *A. sobria* have been implicated as causative agents for human diseases. There are also rare reports of *A. trota* and *A. jandaei* isolation from clinical samples [2,7,8,9,10]. 

*Aeromonas* are responsible for causing skin, soft tissue, urinary tract, and hepatobiliary tract infections, as well as wound infections, peritonitis, pneumonias, and severe bacteremia. In several studies, *Aeromonas*-associated hemolytic uremic syndrome, burn-associated sepsis, and a variety of respiratory infections, including epiglottis, have been reported [2]. However, several studies have shown that in rare cases, *Aeromonas* spp. can be isolated from stool samples of healthy individuals. An association with acute and prolonged diarrhea has also been proven, with reports ranging from 2% to 20% [2,6,7,11,12,13]. The reported rates differ depending on the use of culture methods, species identification methods, and inclusion or exclusion of other diarrheal co-pathogens laboratory investigation alongside *Aeromonas* [14]. There is a wide geographical variation in the reported frequency of isolation of *Aeromonas* spp. and their association with diarrhea. For instance, while the European Union, USA, and Canada report a low incidence of disease caused by *Aeromonas* spp., in developing countries, *Aeromonas* plays a significant role in causing gastroenteritis among children [2,3,7,8,12,13,14,15].

Several factors, including patient age, state of immunocompetence, presence of concomitant diseases, infectious load, and expression of virulence factors, affect the pathogenicity of *Aeromonas* spp. Risk groups include children under the age of five, elderly patients, and immunocompromised patients [8,13,16]. According to the literature, the pathogenic potential of *Aeromonas* results from several virulence factors that allow these bacteria to adhere, invade, and destroy the host’s cells, overcoming its immune response [2,3,5]. Hemolytic toxins include: aerolysin-related cytotoxic enterotoxin (*act*), heat-labile cytotonic enterotoxin (*alt)*, heat-stable cytotonic toxins (*ast)*, hemolysin (*hlyA*) and aerolysin (*aerA*). In addition, the type III secretion system (*TTSS*), polar flagellum (*fla*), lateral flagella (*laf*), elastase (*ela*), and lipase (*lip*) contribute to the pathogenicity of *Aeromonas* [12,17,18,19,20].

Usually, in cases of gastroenteritis, *Aeromonas* is a co-infection, isolated along with other enteric pathogens [21]. Most incidents of acute *Aeromonas*-associated diarrhea are transient and self-limiting [19]. However, dysentery and severe cholera-like diarrhea have been described and may require treatment with antibiotics [8,12,14]. *Aeromonas* spp. strains are universally considered to be resistant to penicillin due to their production of beta-lactamase. They are, however, usually sensitive to aminoglycosides, tetracyclines, chloramphenicol, trimethoprim-sulfamethoxazole, quinolones, and second and third generation cephalosporins [5,8]. 

Although the role of *Aeromonas* spp. in causing gastroenteritis still remains controversial, they have been identified as important pathogens in pediatric practice. There is, however, limited availability of data in the literature regarding such cases in Latvia. 

## 2. Materials and Methods

### 2.1. Sample Collection

Stool samples (n = 1360) were obtained from the Children’s Clinical University Hospital’s Emergency and Observation Department (EAOD) and Infectious Disease Department between 1 January 2020 and 1 October 2021.

In Children’s Clinical University Hospital fecal sample collection for viral antigen testing (Rotavirus, Norovirus and Adenovirus) and for pathogenic bacterial cultures (for *Salmonella*, *Shigella*, and *Yersinia* cultivation in almost all patients and for *Campylobacter* cultivation—mostly in patients with a suspicion of bacterial enterocolitis) is routinely performed in hospitalized patients with suspicion of gastrointestinal tract infection. The target population was patients between 0 to 18 years of age with a preliminary diagnosis of A04/Other bacterial intestinal infections, A09.0/Other and unspecified gastroenteritis and colitis of infectious origin, and A09.9/Gastroenteritis and colitis of unspecified origin (main diagnoses routinely mentioned in the prescriptions for laboratory testing), in whom fecal samples for bacterial culture were collected. The same samples were used for isolation and identification of *Aeromonas* spp. for research purposes, no supplement intervention was required.

In addition, fecal samples (n = 123) from immunocompromised patients were tested for a wide range of bacteria including *Aeromonas* spp. 

Parents of the patients whose stool samples were positive for *Aeromonas* spp. were asked to complete a questionnaire about the symptoms of their child’s disease, their duration and severity, chronic comorbidities and their treatment, vaccination status, and epidemiological factors that could have been associated with infection.

Excluding criteria were repeated episodes of hospitalization with the already known etiology of infectious gastroenteritis and patients who had already received antibiotic therapy before admission (except immunocompromised patients, tested for full bacterial range). 

### 2.2. Isolation and Identification of Aeromonas spp.

The samples were collected using sterile fecal sample containers within 24 h from the moment of hospitalization and delivered to the laboratory immediately or stored first at 2–8 °C for up to 24 h. The samples were then applied on Biolife Italiana CIN agar plates (Typical formula after reconstitution with 1 L of water: Peptone 20.000 g, Yeast extract 2.000 g, Mannitol 20.000 g, Sodium pyruvate 2.000 g, Sodium chloride 1.000 g, Magnesium sulphate 0.010 g, Sodium desoxycholate 0.500 g, Irgasan 0.004 g, Neutral red 0.030 g, Crystal violet 0.001 g, Agar 12.000 g) with selective supplements (Cefsulodin 7.50 mg, Novobiocin 1.25 mg per 500 mL of medium) and incubated in aerobic conditions for 48 h at 28 ± 1 °C [22]. Additionally, the same samples were applied on Biolife CIN agar plates without supplements and incubated in aerobic conditions for 24 h at 35 ± 1 °C (in-house method). Results were then compared between the two methods. Oxidase test was performed with a Becton Dickinson Oxidase Reagent Dropper. Positive colonies were then selected for further identification with the Bruker Maldi-TOF biotyper. 

### 2.3. Antimicrobial Resistance Testing

Antimicrobial resistance of isolates was investigated with the bioMerieux VITEK 2 automated system according to the Clinical and Laboratory Standards Institute’s (CLSI) current guidelines. Positive susceptibility reported to hospital physicians included amikacin (AN), ciprofloxacin (CIP), imipenem (IMI), trimethoprim/sulfamethoxazole (SXT), ceftazidime (CAZ), amoxicillin/clavulanate (AMC), gentamicin (GN), piperacillin/tazobactam (TZP), cefepime (FEP) and cefuroxime (CXM). Strains were stored in a Luria broth: glycerol mixture (80:20) at −80 °C prior to DNA extraction.

The *Aeromonas hydrophila* ATCC^®^ 35654™ reference strain was used for internal quality control and external quality control was provided by Labquality External Quality Assessment (LEQA). *E. coli* ATCC^®^ 25922™ reference strain was used as the quality-control for susceptibility testing.

### 2.4. Detection of Virulence-Associated Genes

Total chromosomal *Aeromonas* spp. DNA was extracted by Analytik Jena innuPREP Bacteria DNA kit according to the manufacturer’s specifications. The presence of 9 genes encoding for putative virulence factors (*ast*, *lip*, *ela*, *act*, *alt*, *aerA*, *hlyA*, *fla*, and *laf*) were identified by polymerase chain reaction (PCR) [20]. Primers synthesized by Metabion International were used for PCR amplification. The reaction was performed with a final volume of 50 μL of Master Mix (5 μL PCR buffer ×10, 2 μL dNTP 2.5 mM, 6 μL MgCL_2_, 1 μL primer (F), 1 μL primer^®^, 1 μL DNS (50 ng/μL), 0.2 μL Taq polymerase (50 U/μL), and 33.8 μL ddH_2_O.) Cycling conditions consisted of an initial 5-min cycle at 95 °C, followed by 35 30-s denaturation cycles at 95 °C. Annealing was performed at 53–62 °C (according to the primer manufacturer’s specification) for 30 s. Elongation was conducted at 72 °C for 1 min and was followed by a final 7-min cycle at 72 °C. Amplification products then were pre-cooled to 15 °C and stored at −20 °C prior to visualization with agarose gel electrophoresis. The final PCR products were then sequenced for further confirmation.

## 3. Statistical Methods

The data were analyzed using IBM SPSS Statistics Version 27. A p-value of more than 0.05 was not considered statistically significant, non-parametric tests were used for further statistical analysis. ANOVA and Spearman’s rho tests were used for correlation analysis.

## 4. Dataset Collection Methods

All retrospective data presented in the study (number of hospitalized patients, performed laboratory tests, etc.) were obtained from Children’s Clinical University Hospital and Laboratory information systems. To estimate a possible impact of COVID-19 pandemic associated restrictions on the size of the study population, the total number of stool tests performed for EAOD and Infectious Diseases department’s patients, and positive bacterial and viral findings in comparable time periods before and during COVID-19 pandemic 1.03.2019–1.03.2020 and 1.03.2020–1.03.2021 were compared.

## 5. Results

*Aeromonas* spp. were isolated in 49 stool samples of patients hospitalized with gastroenteritis and in 1 stool sample of an immunocompromised patient that underwent fecal screening. These positive findings made up 3.6% of all gastroenteritis cases and included four *Aeromonas* species: *A. hydrophila* (12), *A. caviae* (26), *A. veronii* (9), and *A. eucrenophila* (1). In two cases, differentiation between *A. caviae and A. hydrophila* was not possible.

In 25 out of 50 cases, *Aeromonas* spp. colonies grew on both types of media in various culturing conditions. In three cases, colonies appeared only on CIN agar plates that contained a selective supplement and cultivation temperature of 28 ± 1 °C. However, in 22 cases, colonies appeared only on CIN agar plates without the supplement and cultivation temperature of 35 ± 1 °C. In all cases, significant growth was observed after 24 h of incubation. Extension of cultivation to 48 h did not improve the results and led to colony overgrowth. 

The study population with positive *Aeromonas* spp. finding (n = 50) contained 28 male (56%) and 22 female (44%) patients. The median age was 4.56 years (IQR (13.3–1.6); amplitude: 0.2–18). In 29 samples, *Aeromonas* spp. were the only positive finding. However, in the rest they appeared alongside the following clinically significant pathogens: *Campylobacter jejuni* (12) *Salmonella Enteritidis* (6), *Salmonella Typhimurium* (1), and *Yersinia enterocolitica* (3). Norovirus Ag was detected in two stool samples, one sample was positive for Adenovirus Ag, and two for Rotavirus Ag. Among these cases, there were five patients that, in addition to *Aeromonas* spp., were diagnosed with a combination of two other infectious agents. These co-infections were *Campylobacter jejuni/Yersinia enterocolitica* (1), *Campylobacter jejuni/Salmonella Enteritidis* (1)*, Campylobacter jejuni*/Norovirus Ag (2)*,* and *Campylobacter jejuni*/Rotavirus Ag (1) [Figure 1]. Seeing as all of these infections have similar clinical symptoms and co-infection had been identified in these particular fecal samples, it was not always possible to determine the primary infection and the main cause of the disease. 

This study coincided with the COVID-19 pandemic. As a result, the imposed restrictions had a significant impact on the spread of viral infectious diseases in Latvia. This may explain the significantly lower number of patients admitted to the Children’s Clinical University Hospital with gastroenteritis in 2020 compared with the same period in 2019 [Figure 1]. In contrast, however, the proportion of positive bacterial findings was higher [Figure 2].

The median length of hospitalization was 2 days (IQR (3.3–2); amplitude: 0.5–13). Median CRP level on admission was 10.8 mg/L (IQR (3.9–34.6); amplitude: 3.9–49.1), median maximal CRP level during hospitalization was 3.8 mg/L (IQR (3.9–49.1); amplitude: 0.1–317.0). Median blood leucocyte level on admission was 10.2 × 10^3^/µL (IQR (7.1–12.9); amplitude: 3.9–28.1), median maximal blood leucocyte level during hospitalization was 10.2 × 10^3^/µL (IQR (7.4–13.0); amplitude: 3.9–28.1). Change in stool volume, frequency, or consistency was observed in 78% (n = 39) of the study population. Diarrhea was the main indication for stool specimen collection. Blood in stool was observed in 28% of cases (n = 14). Moreover, 38% percent of patients (n = 19) presented with vomiting and 60% (n = 30) reported abdominal pain. Fever of ≥38 °C (axillary or temporal measurement) was reported in 64% of patients (n = 32). The most frequent complication was mesenteric lymphadenopathy (mesadenitis)–reported in 20% of patients (n = 10). [Figure 3].

Antibacterial susceptibility testing was performed in all positive cases [Figure 4]. Aside from expected natural resistance, no significant resistance was detected. The majority of *Aeromonas* spp. isolates were sensitive to amikacin (100% of 50 tested samples), ceftazidime (98% of 49 tested samples), ciprofloxacin (90% of 50 tested samples), cefepime (98% of 48 tested samples), piperacillin/tazobactam (91% of 47 tested samples), gentamicin (78% of 50 tested samples), cefuroxime (87% of 40 tested samples), and trimethoprim/sulfamethoxazole (34% of 49 tested samples). Antibacterial susceptibility results were affected by species-specific test limitation defined by the manufacturer.

Antibiotic therapy was prescribed to 22 of the 50 patients. Mean duration of antibacterial therapy was 5.3 ± 0.4 (amplitude 1–10) days. In majority of cases (n = 12) antibacterial therapy was initiated with ampicillin, and later switched to oral medication—trimethoprim/sulphametoxazol (received in total by eight patients as primary or secondary regimen) or azithromycin (received in total by four patients). Other patients received amoxicillin/clavulanate (n = 5), amoxicillin (n = 3), cefuroxime (n = 2), ceftriaxone (n = 1), and metronidazole (n = 3) as an initial prescription medication or as a secondary oral regimen. 

Only 5 out of the 50 patients (10%) repeatedly attended the emergency department or were re-hospitalized during the course of their disease. One of them continued to have symptoms 2 weeks after the conducted therapy and *A. veronii* presence was detected 15 days after the primary result. Additionally, five randomly-selected patients (10% of positive cases in the study) were tested again 2 weeks after the resolution of their symptoms. None of them had a repeated positive bacterial finding. 

Out of 16 questionnaire respondents, 5 patients had attended school or preschool educational institution, 2 had been abroad, 3 had had a known contact with a person who had diarrhoea, 6 had used unboiled water for drinking (mostly tap water, including filtered), 9 had been swimming (in the sea (n = 2), in a river (n = 3), in a lake (n = 3), in a pool or water park (n = 4)), 11 had had contact with pets, 9 had consumed meat or poultry products (including grilled meat), 9 had consumed milk or egg containing foods (including ice cream, cream, free-range eggs), and 5 had consumed freshwater fish and seafood products (including salted or cold smoked fish and sushi) two weeks or less prior to disease onset. In 10 cases, one or several specific food preparation habits were mentioned by the patients’ parents, including washing of meat before cooking (n = 7), usage of the same board/knife for cutting meat and freshly edible products (n = 3), consumption of vegetables/fruit/berries without prior washing. 

Nine potential virulence factors were identified in this study. The distribution of *ast*, *lip*, *ela*, *act*, *alt*, *aerA*, *hlyA*, *fla*, and *laf* coding genes from *Aeromonas* isolates are presented in Table 1. 

The presence of multiple virulence genes and 35 gene combinations were identified in all of the isolates. However, the pattern varied among *Aeromonas* species, the most prevalent gene was *lip*-78% of all isolates. *Alt* was present in 66%, *ast* in 18%, *ela* in 76%, *act* in 54%, *aerA* in 17%, *hlyA* in 9%, *fla* in 56% and *laf* in 30% of the isolates, respectively. The most common gene combinations were *alt/lip/ela/fla* and *alt/lip/ela/fla/laf,* which presented in 4 out of 50 isolates (8%) each.

Although the virulence pattern of *A. hydrophila* may be higher, no effects on the course of the patient’s disease were observed. Additionally, no statistically significant correlations were found between the bacterial species, C-reactive protein (CRP) levels, the highest recorded fever, and administration and duration of antimicrobial therapy. Correlation with the duration of hospitalization could not be determined, as this was often reduced or limited due the restrictions imposed by the COVID-19 pandemic.

## 6. Discussion

Opinions still differ regarding the inclusion of *Aeromonas* spp. in routine gastroenteritis laboratory diagnostics. Several studies have shown that in rare cases *Aeromonas* spp. can be isolated from stool samples of healthy individuals [7]. Usually, in cases of gastroenteritis, *Aeromonas* is a co-infection, isolated along with other enteric pathogens [21]. At the same time an association with acute and prolonged diarrhea has been proven as well with reports ranging from 2% to 20%. *Aeromonas* are commonly isolated from children’s diarrheal specimens in tropical countries; in turn, association with diarrhea in the European Union has been reported to be as high as 2 to 8% [2,3,12,14,15]. The reported rates differ depending on the use of culture methods, species identification methods, and inclusion or exclusion of other diarrheal co-pathogens’ laboratory investigation alongside *Aeromonas* [14].

Alongside other significant pathogens *Aeromonas* spp. are often included in most of Multiplex molecular panels for gastrointestinal infection [23], highly recommended for differential diagnosis in severe cases of illness. 

However, according to the clinical data and the results of this study (positive findings made up 3.6% of all gastroenteritis cases; 1.2% as a sole pathogen and 1.5% as a co-infection), which are statistically similar to those reported in other EU countries, *Aeromonas* spp. definitely has an important role in pediatric practice and requires appropriate attention and monitoring. The median age of the research population was 4.56 years, which is considered to be a risk group according to the literature. Nevertheless, 20 patients (40%) were over 5 years of age and 15 (30%) were over 10 years of age, demonstrating the significance of the infection in different age groups. 

When comparing the results of the two different cultivation methods, better growth was observed with the in-house method: Biolife CIN agar plates without selective supplement, incubation in aerobic conditions for 24 h at 35 ± 1°. Extension of cultivation to 48 h (according to the standard recommendation) did not improve the results, led to colony overgrowth and could mask the target microorganism with other more rapidly growing bacteria. This may explain the relatively small number of reports from hospitals where investigative methods were based on the standard procedure recommendations. Based on the study results, laboratory isolation methods of *Aeromonas* spp. bacteria require certain modifications and should undergo validation in each investigation performing laboratory. 

Isolation of *Aeromonas* spp. from clinical samples appears to have had a seasonality to it—an increase in cases has been observed in August (21.2% of all annual cases), which may be related to a rise in water temperature in natural reservoirs. 

Due to the social distancing requirements introduced during the COVID-19 pandemic, which had coincided with a significant part of the research timeline, it was impossible to trace the potential transmission of bacterial infections within the population, especially in children’s facilities. However, based on this study, it can be surmised that social distancing and other restrictions did not play a significant role in the general spread of gastroenteritis-causing bacteria [Figure 2]. The majority of patients included in this study did not visit swimming pools, public baths, or spas and had limited opportunity to attend preschool and school due to quarantine measures. Potential food-borne sources of *Aeromonas* spp. infection, such as untreated dairy and raw seafood products, were also not indicated in most questionnaires and still remained unclear. Furthermore, no cases of *Aeromonas* infection transmission between family members were observed during this study. 

There were several limitations to this study. It is possible that there were co-infecting organisms such as other non-tested gastroenteritis causing viral, bacterial or parasitic agents that were not laboratory investigated, but could be responsible for some proportion of the diarrheal symptoms observed. 

In this particular study, it was not scientifically determined whether the presence of *Aeromonas* species or virulent factors affects the symptom severity and duration of gastroenteritis in children. However, this association cannot be completely rejected until studies are conducted in a larger population with normal data distribution. 

Antibacterial susceptibility testing was performed for all positive findings, and no significant resistance except intrinsic specific was detected in majority of cases. However, considering the possible antibiotic resistance features of *Aeromonas* spp. [12,24,25], it would also be reasonable to amend the antimicrobial susceptibility testing protocol for appropriate antibacterial therapy selection in severe cases of infection.

In conclusion, the main potential sources and epidemiological significance of *Aeromonas* infection in Latvia still remain unclear. Therefore, it would be advantageous to repeat the conducted patient surveys during a more representative period without imposed social restrictions.

## Figures and Tables

**Figure 1 children-09-01684-f001:**
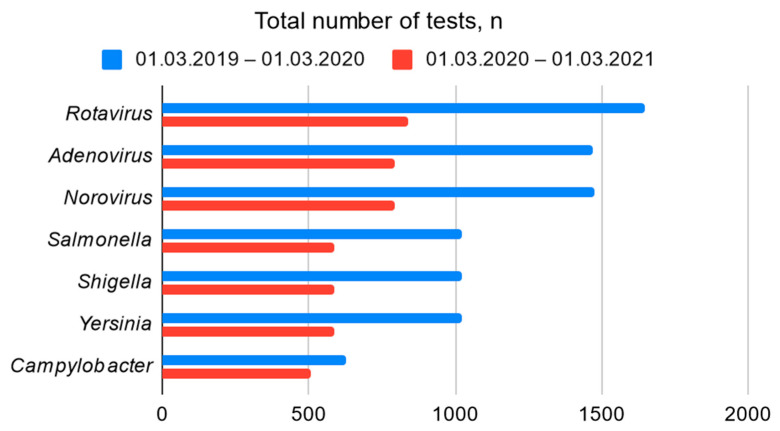
The total number of laboratory tests performed for EAOD and Infectious Diseases department’s patients.

**Figure 2 children-09-01684-f002:**
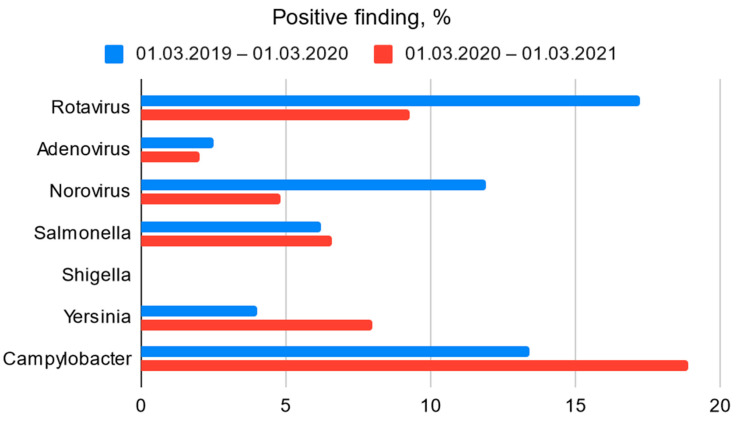
Bacterial and viral (Ag) positive finding among all clinical samples obtained from EAOD and Infectious Diseases department’s patients.

**Figure 3 children-09-01684-f003:**
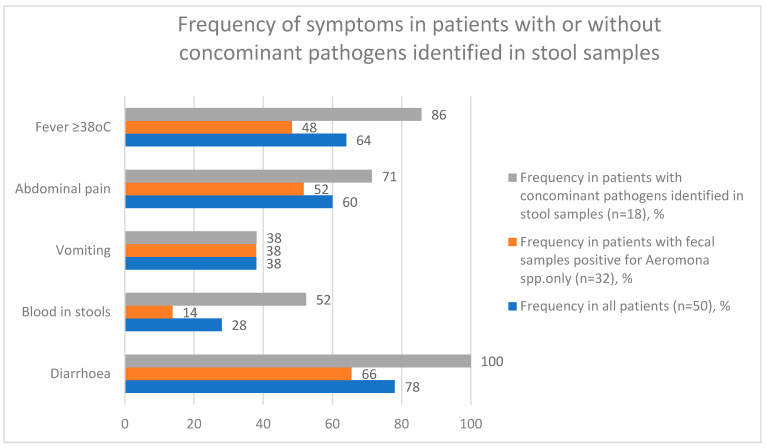
Frequency of symptoms in patients with or without concomitant pathogens identified in stool samples.

**Figure 4 children-09-01684-f004:**
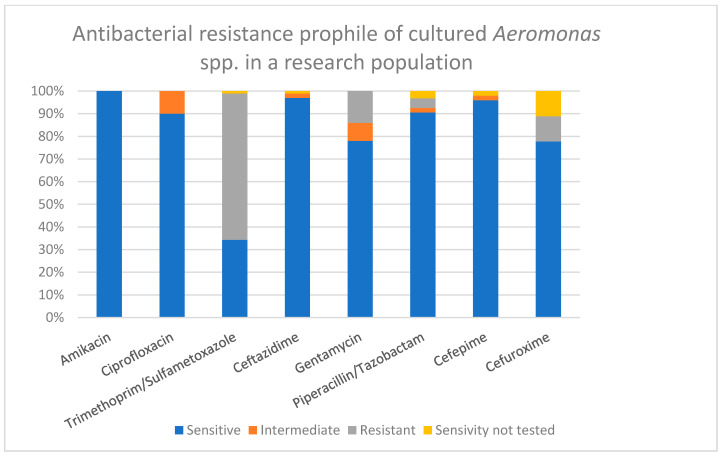
Antibacterial resistance profile of *Aeromonas* spp. isolates, %.

**Table 1 children-09-01684-t001:** Distribution of virulence genes (total strains n, %) among four isolated *Aeromonas* species.

	*alt*	*ast*	*lip*	*ela*	*act*	*aerA*	*hlyA*	*fla*	*laf*
*A. hydrophila*	9 (75%)	8 (66.6%)	12 (100%)	10 (83.3%)	9 (75%)	10 (83.3%)	6 (50%)	6 (50%)	3 (25%)
*A. caviae*	17 (65.4%)	1 (3.85%)	23 (88%)	9 (34.6%)	8 (30.7%)	3 (11.5%)	2 (7.69%)	19 (73%)	9 (34.6%)
*A. eucrenophila*	1 (100%)	0 (0%)	0 (0%)	0 (0%)	1 (100%)	0 (0%)	0 (0%)	0 (0%)	0 (0%)
*A. veronii*	4 (44.4%)	0 (0%)	2 (22.2%)	4 (44.4%)	7 (77.7)	2 (22.2%)	0 (0%)	3 (33.3%)	2 (22.2%)

## Data Availability

The datasets analyzed during the current study are not publicly available due to ethical reasons, but are available from the corresponding author on reasonable request.

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
