# Peer review of "Prevalence of Aeromonas spp. Infection in Pediatric Patients Hospitalized with Gastroenteritis in Latvia between 2020 and 2021"

_children, 2022, doi:10.3390/children9111684_

Round 1
Reviewer 1 Report
The authors isolated Aeromonas from the feces of children in Latvia, and conducted species identification, drug resistance, and search for virulence factors to elucidate the actual situation: during the COVID-19 epidemic, social distance from others had no effect on the duration of hospitalization for Aeromonas infection or bacterial gastroenteritis. The data that the duration of hospitalization was not affected by social distance from others during the COVID-19 pandemic is interesting because it is a characteristic of infections that are not transmitted by humans.
The title is COVID-19 pandemic period, but you need to include previous data as a comparison to see how it relates to the COVID-19 pandemic. If there is no previous data, it would be better to change the title to data for 2020-2021.
In Materials and Methods.
Sample collection, please describe A04, A09.0, and A09.9, which may be a categorization of patients.
In the Isolation and identification of Aeromonas spp., Biolife CIN agar is not a general medium for isolation, so please provide details about the manufacturer and composition.
TTSS is mentioned as a virulence factor in the Introduction, but it seems that no experiments were conducted to detect the gene, so please describe it as an issue for future study. The same for RTX (repeat-in-toxin).
In Results.
Figure 1 and Figure 2 do not describe the methods in the Materials & Methods by which the data were obtained. Please delete or provide details.
In Fig. 3, it would be helpful to see which species of Aeromonas appear more resistant when categorized by Aeromonas species. In particular, readers may be interested in the distribution of TMP/SMX resistance.
Since L140-L164 is an important part of the article, I think it would be better to describe it in a table format for clarity.
For Table 1, if possible, it may be easier to understand the relationship with virulence factors if they are listed by clinical manifestations.
All 'Aeromonas' should be in italics.
Author Response
Point 1. The title is COVID-19 pandemic period, but you need to include previous data as a comparison to see how it relates to the COVID-19 pandemic. If there is no previous data, it would be better to change the title to data for 2020-2021.
Responce to the Point. 1.
Please find requested corrections in the in reviewed manuscript.
Point 2. Sample collection, please describe A04, A09.0, and A09.9, which may be a categorization of patients.
Response to Point 2.
The target population was patients between 0 to 18 years of age with a preliminary diagnosis of A04 / Other bacterial intestinal infections, A09.0 / Other and unspecified gastroenteritis and colitis of infectious origin, and A09.9 / Gastroenteritis and colitis of unspecified origin (main diagnoses routinely mentioned in the prescriptions for laboratory testing), in whom faecal samples for bacterial culture were collected.
Please find corrections in reviewed manuscript.
Point 3. In the Isolation and identification of Aeromonas spp., Biolife CIN agar is not a general medium for isolation, so please provide details about the manufacturer and composition.
Response to Point 3.
The samples were applied on Biolife Italiana CIN agar plates (Typical formula after reconstitution with 1 L of water: Peptone 20.000 g, Yeast extract 2.000 g, Mannitol 20.000 g, Sodium pyruvate 2.000 g, Sodium chloride 1.000 g, Magnesium sulphate 0.010 g, Sodium desoxycholate 0.500 g, Irgasan 0.004 g, Neutral red 0.030 g, Crystal violet 0.001 g, Agar 12.000 g) with selective supplements (Cefsulodin 7.50 mg, Novobiocin 1.25 mg per 500 ml of medium) and incubated in aerobic conditions for 48 hours at 28°±1 C [9].
Point 4. TTSS is mentioned as a virulence factor in the Introduction, but it seems that no experiments were conducted to detect the gene, so please describe it as an issue for future study. The same for RTX (repeat-in-toxin)
Response to Point 4.
The detection of virulence genes most frequently described in the literature was included in the design of the study. Based on the study results, we assume, that it would be advantageous to repeat the study with larger population during a more representative period without imposed social restrictions. Thus, it would be possible to expand the investigation of potential virulence factors as well.
Point. 5.
Figure 1 and Figure 2 do not describe the methods in the Materials & Methods by which the data were obtained. Please delete or provide details.
Responce to point 5.
All retrospective data presented in the study (number of hospitalized patients, performed laboratory tests, etc.) were obtained from Children’s Clinical University Hospital and Laboratory information systems. To estimate a possible impact of COVID-19 pandemic associated restrictions on the size of the study population, we compared the total number of stool tests performed for EAOD and Infectious Diseases department's patients, and positive bacterial and viral findings in comparable time periods before and during COVID-19 pandemic 1.03.2019.-1.03.2020 and 1.03.2020- 1.03.2021.
Point 6.
In Fig. 3, it would be helpful to see which species of Aeromonas appear more resistant when categorized by Aeromonas species. In particular, readers may be interested in the distribution of TMP/SMX resistance.
Responce to Point 6.
In this study, it was not scientifically determined correlation between Aeromonas species and antimicrobial resistance patterns. However, this association cannot be completely rejected until studies are conducted in a larger population with normal data distribution.
According the corrections the Results section, "...aside from expected natural resistance, no significant resistance was detected. The majority of Aeromonas spp. isolates were sensitive to amikacin (100% of 50 tested samples), ceftazidime (98% of 49 tested samples), ciprofloxacin (90% of 50 tested samples), cefepime (98% of 48 tested samples), piperacillin/tazobactam (91% of 47 tested samples), gentamicin (78% of 50 tested samples), cefuroxime (87% of 40 tested samples) and trimethoprim/sulfamethoxazole (35% of 49 tested samples).
Point 7. Since L140-L164 is an important part of the article, I think it would be better to describe it in a table format for clarity.
Responce to Point 7.
Please find an additional explanation in the revised manuscript.
Point 8. For Table 1, if possible, it may be easier to understand the relationship with virulence factors if they are listed by clinical manifestations.
Responce to point 8.
In this particular study, it was not scientifically determined whether the presence of virulent factors affects the symptom severity and duration of gastroenteritis in children. Although the virulence pattern of A. hydrophila may be higher, no effects on the course of the patient's disease were observed. Also no statistically significant correlations were found between the bacterial species, C-reactive protein (CRP) levels, the highest recorded fever, and administration and duration of antimicrobial therapy.
Point 9. All 'Aeromonas' should be in italics.
Corrected
Reviewer 2 Report
1) Aeromonas - in italic. See all article: Title, abstract, ...........table 1 and references. Please review.
2) spp. - not italic. See all article
3) Keywords: Aeromonas, gastroenteritis, pediatric, virulence, antimicrobial resistance
4) Introduction
see Aeromonas and spp.
5) M&M ok
6) Results
- line - 144 - A. caviae
- Figure 1 and 2 - italic Salmonella, Shigella, Yersinia , Campylobacter
- Figure 3 - Inclusion the complete names of the antimicrobials beside or below of the abbreviations.
- lines 187-190 - Why ..."ceftazidime (98% of 49 tested samples), cefepime ( 48 tested samples), piperacillin/tazobactam (91% of 47 tested samples), gentamicin (78% of 50 tested samples), and cefuroxime (87% of 40 tested samples"..... were not done in the 50 tested samples?
And about antimicrobial resistance profile of IMI and AMC. It is better to remove from the figure 3 and to add in the text.
- Table 1 - please, redo and remove the lines.
7) Discussion
- line 222- Why did you do two different cultivation methods?
- lines 223 and 224 - I think it is better to rewrite lines 223 and 224. Discuss which cultivation method was the best and advise the use of.
8) References- see italic
Author Response
Point 1.-5., 8.
Please find requested corrections in reviewed manuscript.
Point 6.
Lines 187-190 - Why ..."ceftazidime (98% of 49 tested samples), cefepime ( 48 tested samples), piperacillin/tazobactam (91% of 47 tested samples), gentamicin (78% of 50 tested samples), and cefuroxime (87% of 40 tested samples"..... were not done in the 50 tested samples?
Responce to Point 6.
VITEK 2 automated system was used for antibacterial susceptibility testing and, unfortunately, results were affected by species-specific test limitation defined by the manufacturer. We added the explanation in the Result section.
Point 7.
- line 222- Why did you do two different cultivation methods?
- lines 223 and 224 - I think it is better to rewrite lines 223 and 224. Discuss which cultivation method was the best and advise the use of.
Responce to Point 7.
Since there is no rutine laboratory procedures for Aeromonas isolation from faecal samples in Latvia, we started pilot study based on general recommendations (Clinical Microbiology Procedures Handbook, American Society for Microbiology, Third Edition, Vol.1.)., but it was noticed, that reference strain growth was not as good as expected. So, we made changes in media composition and incubation conditions, obtaining a significantly better result. However, we avoid recommending our in-house method, as the results may be manufacturer of culture media depending. We suggest isolation method validation in each investigation performing laboratory instead.
Reviewer 3 Report
In general, the manuscript is well written and comprise interesting data, but everything in the paper focusses on Aeromonas as a cause of infecctions. However, nothing is shown about the impact of Aeromonas for the infection. Can the authors exclude that other bacteria then Aeromonas were causing the infection? Furthermore, the isoaltes recovers seem to be very diverse - I could not find any association between the infection of the person, the species and/or the occurrence of a specific virulence factor... so I was wondering why the author argumented so strictly in the direction of a pathogenicity. It is the opinion of this reviewer, that the data will not support these agumentation line. The Paper should be modified in a way to indicate what was found, to derive out of the data potential associations and hen to specify what kind of Aeromonas were found ... Are more judged discussion would be beneficial for the manuscript.
Author Response
Responce to Review report.
During the study fecal sample collection for viral antigen testing (Rotavirus, Norovirus and Adenovirus) and for pathogenic bacterial cultures (for Salmonella, Shigella, Yersinia and Campylobacter) was routinely performed in all hospitalized patients with suspicion of gastrointestinal tract infection. Excluding criteria were repeated episode of hospitalization with the already known etiology of enfectious gastroenteritis and patients who had already received antibiotic therapy before admission (except immunocompromised patients, tested for full bacterial range). According to the clinical data of our patients and the results of this study (positive findings made up 3.6% of all gastroenteritis cases; 1.2% as a only one pathogen and 1.5% as a co-infection) which are statistically similar to those reported in other EU countries, Aeromonas spp. requires appropriate attention and monitoring in paediatric practise.
We also complemented the manuscript in Results and Discussion sections for better clarity.
Reviewer 4 Report
The authors have mentioned in Abstract that “The aim of the study was to describe potential sources, prevalence of infection, associated virulence factors and antimicrobial resistance of Aeromonas spp. isolated from fecal samples”. In the study, authors have not described the potential sources of infection.
In Abstract “Aeromonas spp. were isolated in 50 stool samples, positive findings made up 3.6% of all gastroenteritis cases and included” The meaning is not clear.
In Abstract “The study population contained 28 male 38 Children (56%) and 22 female (44%) patients; median age was 4.56 years”. Is this the study population or the Aeromonas infected children?
From the study design, it is not clear whether Aeromonas was isolated as a sole pathogen in the 50 patients. Moreover, this organism may also transiently colonize the gastrointestinal tract; although, it is not considered part of the normal intestinal flora, therefore the other viral or bacterial pathogens may be involved in diarrhea or some other reason of diarrhea (Antibiotic associated diarrhea). Please make it clear in the methodology.
Author Response
Point 1. The authors have mentioned in Abstract that “The aim of the study was to describe potential sources, prevalence of infection, associated virulence factors and antimicrobial resistance of Aeromonas spp. isolated from fecal samples”. In the study, authors have not described the potential sources of infection.
Response for Point 1.
As it was indicated in Discussion section "At the conclusion of this study, the main potential sources and epidemiological significance of Aeromonas infection in Latvia still remained unclear". Please also find the supplement information it the reviewed manuscript.
Point 2.
In Abstract "Aeromonas spp. were isolated in 50 stool samples, positive findings made up 3.6% of all gastroenteritis cases and included” The meaning is not clear.
Response for Point 2.
1360 stool samples were obtained and tested for Aeromonas presence in the study. 50 samples were positive for Aeromonas infection and made up 3.6% of all samples / all study population. We also added some information in materials and methods section.
Point 3.
In Abstract “The study population contained 28 male 38 Children (56%) and 22 female (44%) patients; median age was 4.56 years”. Is this the study population or the Aeromonas infected children?
Response for Point 3.
The study population (n=1360) was pediatric patients between 0 to 18 years of age. The study population with positive Aeromonas spp. finding (n=50) contained 28 male (56%) and 22 female (44%) patients. Please find the corrected information it the reviewed manuscript.
Point 4.
From the study design, it is not clear whether Aeromonas was isolated as a sole pathogen in the 50 patients. Moreover, this organism may also transiently colonize the gastrointestinal tract; although, it is not considered part of the normal intestinal flora, therefore the other viral or bacterial pathogens may be involved in diarrhea or some other reason of diarrhea (Antibiotic associated diarrhea). Please make it clear in the methodology.
Response for Point 4.
As it was indicated in Results section, "In 29 samples, Aeromonas spp. were the only positive finding. However, in the rest they appeared alongside the following clinically significant pathogens."
We also put some additional information about the laboratory investigation (viral / bacterial) performed alongside with Aeromonas spp. testing in the reviewed manuscript.
Round 2
Reviewer 3 Report
The revision conducted improved the quality of the manuscript
Reviewer 4 Report
The authors have improved the manuscript and can be accepted for Publication.